# Assessment of knowledge, attitude and practice on first aid management of choking and associated factors among kindergarten teachers in Addis Ababa governmental schools, Addis Ababa, Ethiopia. A cross-sectional institution-based study

Ali Maalim Issack[1], Tilahun Jiru[2], Andualem Wubetie Aniley[2]*

1 Department of Nursing, College of Medicine and Health Sciences, Jigjiga University, Jigjiga, Ethiopia,
2 Department of Emergency Medicine, School of Medicine, College of Health Sciences, Addis Ababa University, Addis Ababa, Ethiopia

* andualemwubete@yahoo.com

## Abstract

### Background

Choking refers to a blockage of upper airways by food or other objects resulting in interruption of breathing. It is a medical emergency that needs immediate action by anyone near by the victim to save life. Chocking is a major cause of illness and death in the pediatric population under the age of 5 years. Children at this age spent more time in their school and are at high risk during their feeding and playing. Immediate provision of first aid in response to choking by a preschool teacher will help to decrease the risk of developing life-threatening complications, length of hospital stays, the cost of treatment, and death.

### Methods

Institutional-based cross-sectional study design was applied to the study area using pre-tested, structured, and self-administered questionnaires. The collected data were analyzed using SPSS version 25. Multiple logistic regression analysis was used to identify factors associated with Knowledge, attitude, and practice of kindergarten teachers towards first aid management of choking.

### Results

A total of 224 Kindergarten teachers were involved in the study with a response rate of 95%. Only eighty-three (37%) of them were knowledgeable and 97 (43.3%) have faced a choked child in the school compound. Of these, only 42 (43.2%) had provided first aid to the victim. Most of the respondents 95.1% had a positive attitude towards choking first aid and 57.1% of them agreed that choking needs immediate management. Multiple logistic regression analysis showed that Kindergarten teachers with the previous first aid training were 2.9

**Data Availability Statement:** All relevant data are within the paper and its Supporting Information files.

**Funding:** The authors received no specific funding for this work.

**Competing interests:** No authors have competing interests

times more knowledgeable than those kindergarten teachers without previous first aid training (AOR: 2.902, 95% CI: 1.612, 5.227)

## Conclusions

The level of knowledge and skills for providing first aid for choking children among kindergarten teachers is low. There is a need for urgent intervention to train teachers regarding the provision of first aid for choking children.

## Background

Choking refers to a blockage of upper airways by food or other foreign bodies resulting in interruption of breathing. It is an actual life-threatening emergency that needs immediate action by anyone nearby the victim to save a life [1]. Foreign bodies in the upper airway can cause acute obstruction leading to the onset of respiratory distress. It is common in children, who ingest objects as they pick up everything and place it in their mouths. Chocking is among the most common cause of injury-related morbidity and mortality especially under the age of 4 years [2, 3]. It is the major cause of illness and death in the pediatric population. Preschool-aged children are at higher risk for the choking incident because their behavior predisposes them to it [4]. The choking incident happens more frequently in young children as they play with the small item and put them in their mouths that may easily lodge into their airways resulting in an obstruction. If the air passage is not cleared, it can lead to loss of consciousness within 3–5 minutes. In worse cases, it results in hypoxia, and brain ischemia leads to death within few minutes [5]. Even if there are variations, the most common objects for choking on children are food, coins, toys, and balloons which commonly occur during feeding and playing [3]. Common signs and symptoms of choking include, cough, difficulty to breathe or talk, grasping the throat, and cyanotic appearance [6]. Choking can be managed with basic first aid skills applying a combination of abdominal thrust and back blow for children over one year. Abdominal thrust commonly called the Heimlich maneuver is a more effective technique of intervention to remove foreign body obstruction from the airway [1, 6, 7].

Studies showed that children are at increased threat for choking during school hours. Immediate provision of basic life support interventions by their teachers will prevent life-threatening complications. Kindergarten(KG) teachers are the major caregivers and first line of protection for KG school children replacing the role of parents [5, 8]. Educating and qualifying teachers in basic life support will help improve the survival of children for sudden out-of-hospital cardiac arrest. They will also teach children about first aid at the early stage of their life to know the preventive mechanism. Training teachers is easy to become competent in Basic Life Support (BLS) [9, 10]. Life-threatening conditions have a greater chance of happening at the educational institution since learners use more time at school and liable for accidents. Majority of an accident happening at preschool need immediate prehospital care intervention. As preschool teachers being the first respondent in the preschool setting, they need to be trained on first aid care to decrease illness and death related to an accident at the school [11].

Different teaching mechanisms have been proposed by organizations to educate about the prevention and management of choking. Different organizations like Red cross [12], American Heart Association [13], European Resuscitation Council [7] have developed guidelines for first aid management procedures including choking first aid. Many countries have used these

guidelines to train their citizens for the prevention and treatment of choking first aid. Children are given special attention because of their susceptibility to choking [14].

Studies conducted in Iran, China, and Turkey showed that preschool teachers had inadequate knowledge and skills to handle an accident in a school setting [15–17]. A study conducted in Iraq reviled that most of the teachers have a low to moderate level of knowledge regarding chocking first aid management which was significantly associated with teaching years of experience. The study also showed that most of the teachers have good attitudes significantly associated with their age, marital status, educational background, and levels of experience [5]. Another study done in South Africa Cape town showed that only 12.1% of teachers have adequate knowledge of first aid management [18].

Previous studies done in Ethiopia on KAP of first aid found that KG teachers have inadequate knowledge and skill towards first aid [19, 20]. A study done in the Lideta sub-city of Addis Ababa showed that 40% of the participants have scored above the mean of knowledge questions and only 37.6% of them have good knowledge towards chocking first aid. Twenty-eight percent of preschool teachers identified choking students in the school setting [19].

As far as investigators' knowledge, there is no study conducted in Ethiopia on choking first aid. Therefore, this study aimed to explore kindergarten school teachers' choking first aid knowledge, attitude, and practice at government schools in Addis Ababa, Ethiopia.

## Methods

### Study setting and period

This study was conducted in selected public kindergarten schools in Addis Ababa city from March-1 to April 31, 2019. Addis Ababa is the capital city of Ethiopia which covers an area of about 520.14 km$^2$. It has ten sub-cities that encompass 116 districts. According to the National Population and Housing Census of Ethiopia projection figures in 2011, the total population of Addis Ababa is 2,980,001 with a proportion of male 47.64% and female 52.36% [21].

There are a total of 239 government kindergarten(KG) schools in Addis Ababa city with 1520 female and 43 male employed teachers [22]. The study was conducted at government KG schools in Bole, Gulele and kirkos sub-cities of Addis Ababa, Ethiopia.

### Study design

Cross-sectional institution-based study design was conducted. The study was conducted in selected sub-cities of Addis Ababa governmental KG schools.

### Sampling

A multistage random sampling technique was used. Based on the World Health Organization's (WHO) 30% of minimum cluster size recommendation for resource-limited countries [23, 24], three sub-cities namely; Bole, Gulele and Kirkos sub-cities were selected with a simple random sampling method from ten sub-cities found in Addis Ababa city. Similarly, 22 districts from 37 were selected, and finally study was conducted on 22 schools from 50 in the districts of 3 sub citiess' kindergarten schools (**Fig 1**).

The sample size was calculated using a single population proportion formula n = $(Z\alpha/2)^2$ *p (1-p) /d2, by taking the assumptions as Z$\alpha$/2 = 1.96 (standard normal value corresponding to 95% level of confidence), p = 0.5 (estimate of prevalence for KAP of chocking first aid to be 50%, since there is no similar study conducted in the study area), and d (margin of error) =

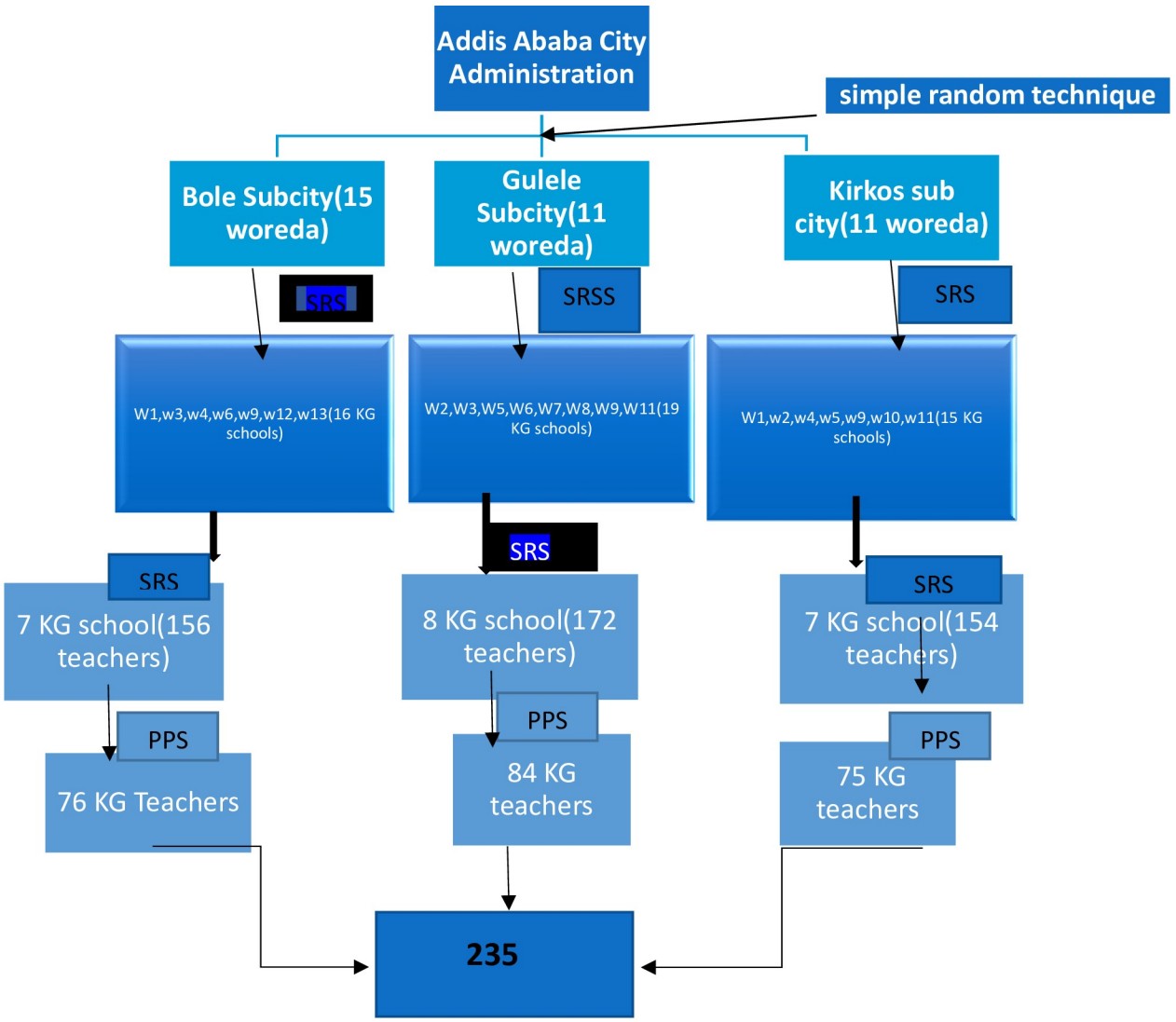

KG-Kindergarten

SRS- simple random sampling

 PPS- probability to proportional size allocation

W-Woreda (District)

**Fig 1. Schematic presentation of sampling procedure for kindergarten teachers among selected government schools in Addis Ababa Ethiopia, 2019.**

5%. The finite population correction formula was used for the study population less than 10,000 since the total population was 482 from 22 selected KG schools. After adjusting 10% for possible nonresponse rate, the final sample size was 235.

## Data collection

Data were collected using pretested, structured self-administer questionnaire which consisted of socio-demographic information, knowledge questions, Attitude questions and Practice question of kindergarten teachers on first aid management of choking.

A questionnaire from previous investigations was modified and used [5, 7–9, 19, 25, 26]. The questionnaire has four parts;

**Part one:** consisted of socio-demographic characteristics of the teachers including age, sex, marital status, educational level, service year and previous first aid training.

**Part two:** of the questionnaire was designed to assess knowledge of teachers about choking first aid which has 11 multiple-choice questions. Each question has one correct answer, 1 correct and 0 incorrect answers with the minimum and maximum possible scores of 0 and 11Teachers scored the mean value and above of knowledge questions were considered to be knowledgeable.

**Part three:** consisted of seven questions focused to assess the attitude of KG teachers towards choking first aid using a five-point Likert scale with the value as follow; 1 score for strongly dis agree, 2 for disagree,3 for not sure, 4 for agree and 5 for strongly agree. Negative items were reverse coded and the answers to show low to a high level for the questions were negatively worded (1-strongly agree, 5 strongly disagree). Participants who responded agree and strongly agree have taken as positive(1) and those disagree and strongly disagree were negative attitude(0). The score mean and above for attitude questions was considered as positive attitude and below mean was negative attitude.

**Part four:** consists of 12 case-based multiple questions structured to assess teachers' practice for the provision of chocking first aid. KG teachers who scored 80% and above of practice questions have good practice for choking first aid. The validity of the questionnaire was assessed and evaluated by experts specialized in the field related to the present study. The reliability of Cronbach's Alpha test was = 0.802

The questionnaire was prepared in English language and translated to local Amharic language by experts and then translated back to the English version to ensure its consistency.

Four-degree holder nurses were collected the data who have trained for two days on clarification of some terms questionnaire and aim of the study. Concerning the need for strict confidentiality of respondents' information, timely collection and reorganization of the collected data from respective kindergartens. KG teachers who were ill, on maternity leave, annual leave, and study leave, were excluded from the study.

## Data quality management

Data were guaranteed during collection, coding, entry, and analysis for appropriate quality assurance. The questionnaire was pre-tested on 5% of the calculated sample size out of the study area before the actual data collection, on a similar population. Some modifications were made to the questionnaires according to the participants' recommendations and the average time taken to fill the questionnaire was 10 minutes.

Supervisors and principal investigators have checked data collectors on how they were administering questionnaires to the study participants and completeness of data on a daily basis and feedback was given. Codes was given to the questionnaires during the data collection. Consequently, any problem encountered was discussed among the survey team and was solved as soon as possible. The collected were coded and entered Epi-Data version 7.0, cleaned, and finally transferred to SPSS version 25 for further analysis.

## Data analysis

KG teachers' Knowledge of first aid provision for choking was assessed using eleven questions. The questions were dichotomized to knowledgeable and not-knowledgeable. A score of mean value and above it was considered as knowledgeable, while less than the mean value taken as not-knowledgeable. Knowledge was taken as a dependent variable and independent variables were sex and age of participants, year of experience, marital status and previous first aid training. The variables were taken from previously studied literatures. Teachers' attitude towards first aid management for chocked child was assessed by using a five-point Likert scale. Strongly agree-5, agree-4, neutral -3, disagree 2 and strongly disagree-1. A score of 5 and 4 were given for positive attitude while 2 and 1 were for negative attitude and the order of scoring for negative statements was reversed. Then, the score was dichotomized into positive and negative attitude for each question. Practice of teachers were described and summarized using descriptive statics.

Bivariate and Multivariate logistic regression analysis were used to identify factors associated with the knowledge, attitude, and practice of kindergarten teachers on first aid management of choking. Univariate logistic regression analysis was used to identify potential associated factors between dependent and independent variables. To control for the potential confounders, a multiple logistic regression model with backward selection was used. For those variables with level statistically significant (P < 0.25), on univariate analysis were entered jointly into a multivariate logistic regression. All statistical tests were two-tailed, and the significance level was declared at p<0.05 with a 95% confidence level. Regression was applied to assess the association between dependent and independent variables

## Ethics statement

Approved ethical clearance letter was obtained from Addis Ababa University health science college, department of emergency medicine ethical review committee. An official letter was written to Addis Ababa City administration Education bureau to grant permission to undertake a study. Permission was obtained from selected Sub City education bureau. Participants were informed verbally and those who were not volunteers had been permitted not to participate in the study. Informed written consent was obtained from respondents who had participated in the study. Confidentiality was maintained throughout the study by assuring that any information will never be passed to any third party or institution without their agreement.

## Operational definition

Choking first aid–is the immediate care given to choking person until full medical treatment is available.

Adequate knowledge–a participant who scored a mean and above for knowledge questions.

Inadequate knowledge–a participant who scored below mean for knowledge questions.

Positive Attitude- a participant who answered agree and strongly agree for attitude questions and sored mean or above.

Negative attitude–a participant who answered disagree and strongly disagree for attitude questions and scored below mean.

Good Practice–a participant who scored 80%and above of practical questions (according to AHA pediatric basic life support and advanced life support 2020)

Poor practice–a participant who scored below 80%of practical questions (AHA, PBLS and PALS 2020)

## Results

### Socio-demographic characteristics

Out of 235 participants, 224 were correctly responded to the provided questions with a response rate of 95%. Most respondents,211(94.2%) were females with a female to male ratio of 16:1. The average age of the respondents was 30.79±7.26years. Concerning the educational level of the kindergarten teachers, the majority (152, 67.9%) had a certificate level(secondary school complete with additional two years teachers' education training); which is a minimum requirement to be a KG teacher in Ethiopia and 2 (0.9%) were Bachelor Degree holders. Regarding the years of experience, 98 (43.8%) had less than five years of experience. The study revealed that the majority (122, 54.5%) had not taken previous first aid training (Table 1).

### Knowledge of kindergarten teachers on first aid management of a choking child

Of all respondents, only 37% of KG teachers scored above the mean value of knowledge questions. One hundred thirty (58%) of participants have heard about first aid provision for a choking child. Thirty-nine of them (29%) heard it from health professionals and the rest from family members (27.6%) and media (26.1%). Majority of respondents,117(52.2%) did not know how to give first aid for a choking child. From those who know,40 (37.4%) learned it from health professionals and 3 (2.8%) from previous studies.

One hundred twenty (53.6%) of participated teachers were knowledgeable about the universal sign of choking. Eighty-four (37.5%) of teachers know risk factors for choking including improper chewing of food, immature molars, running and playing with food in their mouth and adventurous nature of preschool children.

Majority (50.9%) know that coin has a potential for causing a choking hazard while whole grapes (16.5%) and popcorn (4.9%) are the least to cause it. One hundred fifty-nine participants (71%) responded two minutes as a golden time for providing first aid while (6.7%) mentioned one hour. Concerning symptoms of complete airway obstruction, most of the

**Table 1. Socio-demographic characteristics of kindergarten teachers among government schools in Addis Ababa Ethiopia, 2019.**

|  | variable | Frequency | Percentage |
|---|---|---|---|
| sex | Male | 13 | 5.8 |
|  | Female | 211 | 94.2 |
| Marital status | Married | 111 | 49.6 |
|  | Single | 110 | 49.1 |
|  | Divorced | 3 | 3.1 |
| Level of education | Certificate | 152 | 67.9 |
|  | Diploma | 70 | 31.3 |
|  | Degree | 2 | 0.9 |
| Years of experience | < 1 year | 37 | 16.5 |
|  | 1–5 years | 98 | 43.8 |
|  | 6–10 years | 58 | 25.9 |
|  | >10 years | 31 | 13.8 |
| Age Groups | 20–24 | 42 | 18.8 |
|  | 25–29 | 67 | 29.9 |
|  | 30–34 | 60 | 26.8 |
|  | 35–39 | 28 | 12.5 |
|  | 40 and older | 27 | 12.1 |
| Previous first aid training | Yes | 102 | 45.5 |
|  | No | 122 | 54.5 |

**Table 2. Respondent's answers frequency and percent for specific questions on knowledge aassessment in Addis Ababa, Ethiopia, 2019.**

| Questions | Yes | | No | |
|---|---|---|---|---|
| | Frequency | Percentage | Frequency | Percentage |
| The universal sign of choking | 120 | 53.6 | 104 | 46.4 |
| Factors led to choking among preschoolers | 84 | 37.5 | 140 | 62.5 |
| Potential choking hazard item | 29 | 12.9 | 195 | 87.1 |
| Golden time for providing choking first aid | 9 | 4 | 215 | 96 |
| Symptoms of complete airway obstruction | 126 | 56.3 | 98 | 43.8 |
| Symptoms of partial airway obstruction | 59 | 26.3 | 165 | 73.7 |
| Choking induced by aspiration of fluids | 125 | 55.8 | 99 | 44.2 |

participants 126 (56.3%) were knowledgeable and described it as an inability to produce sound and cough (Table 2).

## Attitude of kindergarten teachers towards first aid management of choking

From the total participants, Majority (95.1%) of them scored the mean and above of attitude questions considered to have positive attitude towards providing first aid for a chocking child. One hundred twenty-eight (57.1%) and 66(29.5%) of participants agreed and strongly agreed that choking needs immediate management respectively. Similarly, most of the respondents, 153(68.3%) strongly agree that everybody should know about first aid management of choking. Ninety-nine (44.2%) strongly disagree that choking causes death or life-threatening condition if not treated. The majority,113(50.4%) agree that it was possible to manage choking at school without taking the victim to the health institution. Seventy-five (33.5%) strongly disagree that they should sweep their fingers blindly into the throat of a choked victim and 24 (10.7%) strongly agree to it. Majority of respondents 106 (47.3%) agreed not to provide first aid without proper knowledge on how to do it (Table 3).

## Practice of kindergarten teachers on first aid management of choking

Majority of the participants, 123(54.9%) witnessed a choking episode outside the kindergarten. Among these, 63 (51.2%) had provided first aid. Ninety-seven (43.3%) were encountered a choking episode in kindergarten. Of these, 55 (56.7%) of them did not provide first aid. The reason for not providing first aid was lack of knowledge 79 (68.7%), fear of complications 19 (16.5%), fear of medico-legal issues 13 (11.3%) and fear of communicable disease transmission 4 (3.47%).

**Table 3. Attitude on choking first aid management among government kindergarten school teachers in Addis Ababa Ethiopia, 2019.**

| Question items | Strongly Agree | Agree | Neutral | Disagree | Strongly disagree |
|---|---|---|---|---|---|
| Choking should need immediate management | 66(29.5%) | 128 (57.1%) | 0 | 23 (10.3%) | 7(3.1%) |
| Everyone should know about first aid management of choking | 153(68.3%) | 54(24.1%) | 0 | 8(3.6%) | 9(4%) |
| Choking does not cause death/life threatening condition even if not treated | 12(5.4%) | 31(13.8%) | 0 | 82 (36.6%) | 99(44.2%) |
| It is possible to manage choking at school without taking a victim to the health hospital | 49(21.9%) | 113 (50.4%) | 0 | 43 (19.2%) | 19(8.5%) |
| We should sweep our fingers blindly into the throat of choked victim & take victim to health institution | 24(10.7%) | 57(25.4%) | 0 | 68 (30.4%) | 75(33.5%) |
| You must not provide choking first aid without knowledge | 35(15.6%) | 106 (47.3%) | 0 | 56(25%) | 27(12.1%) |
| If choking first aid is not given within golden time, it may lead to death | 126(56.3%) | 76(33.9%) | 0 | 16(7.1%) | 7(2.7%) |

Some participants 91(40.6%) prefer to give a glass of water for a choking child while 50 (22.3%) would hit the back of the neck and 167 (74.6%) send the victim to a health institution.

Seventy-eight (34.8%) of the respondents did not know the site of the body to provide choking first aid for complete airway obstruction while 52 (23.2%) would tap just below the neck and 29 (12.9%) would tap between shoulder blades and the base of the ribs. Ninety-eight participants (43.8%) would give a glass of water and call for an ambulance if they faced a child choking and coughing with complete airway obstruction and visible foreign body. And, seventy-six (33.9%) would contact the responsible school authority if the child lost consciousness and become breathless.

Based on assessed practical questions for complete airway obstruction management 67 (29.6%) would do a finger sweep, 152 (67.9%) would give water and 4 (1.8%) would give a piece of food (Table 4). Overall,97.8% of study participants had scored below 80% of practical questions they are considered to be poor practice towards first aid management of choking.

**Table 4. Respondent's answer for specific question on practice assessment among govenment kindergaten school teachers in Addis Ababa, Ethiopia, 2019.**

| Questions | Response | Frequency | Percentage |
|---|---|---|---|
| Faced choking victim outside school (n = 224) | Yes | 123 | 54.9 |
| | No | 101 | 45.1 |
| Given first aid n = 123 | Yes | 63 | 51.2 |
| | No | 60 | 48.8 |
| Faced a choking child in the school n = 224 | Yes | 97 | 43.3 |
| | No | 127 | 56.7 |
| Action taken when faced child choking with complete Airway obstruction, object no visible | Giving a glass of water | 91 | 40.6 |
| | Do finger sweep to identify & remove object | 16 | 7.1 |
| | Hitting at the back of neck | 50 | 22.3 |
| | Abdominal thrust | 24 | 10.7 |
| | Slapping at the back | 36 | 16.1 |
| | Don't know what to do | 7 | 3.1 |
| Action taken when faced a child choking, develop talking and breathing difficulty with visible and accessible foreign body. | Taking to health institution | 51 | 22.8 |
| | Notifying the school director | 35 | 15.6 |
| | Remove foreign object | 26 | 11.6 |
| | Hitting at the back of neck | 54 | 24.1 |
| | Giving a sip of water | 42 | 18.8 |
| | Abdominal thrust | 14 | 6.3 |
| | Chest thrust | 2 | 0.9 |
| Action taken when child is choking and coughing | Slap at the back | 52 | 23.2 |
| | Give a glass of water & call EMS | 98 | 43.8 |
| | Abdominal thrust | 38 | 17 |
| | Encourage a child to cough | 22 | 9.8 |
| | Chest thrust | 14 | 6.3 |
| Child choked, became breathlessness and unconsciousness | Contacted responsible school authority | 76 | 33.9 |
| | Slapped at the back | 54 | 24.1 |
| | Given two rescue breath and do CPR | 59 | 26.3 |
| | Begun to do CPR | 11 | 4.9 |
| | Don't know what to do | 22 | 9.8 |
| | Hitting at the back of neck | 1 | 0.45 |
| | Did finger sweep | 1 | 0.45 |

## Factors affecting knowledge of kindergarten teachers toward first aid management of choking

Binary and multiple logistic regression analysis were done to analyze factors associated with knowledge of providing choking for first aid. On the binary logistic regression analysis, sex, age, marital status, years of experience and previous first aid training were all associated with knowledge of first aid management for a chocked child. Previous first aid training experience was significantly associated with teachers' knowledge of choking first aid management (AOR: 2.902, 95% CI: 1.612, 5.227, P ≤0.05). Demographic data of KG teachers has no association with their attitude and practice of choking firs aid (Table 5).

## Discussion

Adequate knowledge and skills of kindergarten teachers can help in the prevention and reduction of morbidity and mortality related to choking episodes in the KG school compound. The teachers play a major role in the management of these emergencies. Training basic life support including choking first aid will improve the survival of children related to chocking in the school environment [9].

The knowledge and skill of kindergarten teachers towards choking first aid management for choked children in this study was inadequate. Only 37% of KG teachers scored above the mean value of knowledge questions. The absence of compulsory first aid management during kindergarten teachers' training in Ethiopia could be the reason for this finding. The finding of this study was in line with the studies conducted in other countries; In Iraq, most of teachers had poor knowledge [5] and in India, 6 out of 146 teachers have good knowledge about first aid provision of a chocking child [8] and in China, only 30.1% answered correctly for first aid management of chocking [16]. A study done in South Africa Cape town also showed that only 12.1% of teachers have adequate knowledge on first aid management [18]. But a study conducted in Egypt was higher than our finding in that majority of teachers were knowledgeable

**Table 5.** Binary and multiple logistic regression analysis of selected factors affecting knowledge on first aid management of choking among government kindergarten school teachers in Addis Ababa, Ethiopia, 2019.

| Variable | | knowledge level | | Linear regression (95% CI) | | | |
|---|---|---|---|---|---|---|---|
| | | Adequate | inadequate | COR(p<0.25) | p- value | AOR(P<0.05) | p- value |
| | | Freq. (%) | Freq. (%) | | | | |
| Sex | Male | 7(3.1%) | 6(2.7%) | 1.729(0.562–5.325) | 0.340 | 2.699(0.768–9.487) | 0.122 |
| | Female | 76(33.9%) | 135(60%) | 1 | | 1 | |
| Age | 20–24 | 10(23.8%) | 32(76.2%) | 1 | | 1 | |
| | 25–29 | 26(38.8%) | 41(61.2%) | 2.03(0.855–4.812) | 0.108* | 2.117(0.780–5.747) | .141 |
| | 30–34 | 25(41.7%) | 35(58.3%) | 2.28(0.952–5.489) | 0.06* | 2.224(0.790–6.262) | 0.13 |
| | 35–39 | 8(28.6%) | 20(71.4%) | 1.28(0.433–3.787) | 0.656 | 1.278(0.373–4.378) | 0.696 |
| | 40 and older | 14(51.9%) | 13(48.1%) | 3.45(1.222–9.712) | 0.019** | 2.845(0.758–10.674) | 0.121 |
| Experience | < 1 yrs. | 14(37.8%) | 23(62.2%) | 1 | | 1 | |
| | 1-5yrs. | 37(37.8%) | 61(62.2%) | 0.440(0.166–1.165) | 0.098* | 0.611(0.238–1.572) | 0.307 |
| | >5–10 yrs. | 23(39.7%) | 35(60.3%) | 0.438(0.193–0.997) | 0.049** | 0.513(0.182–1.446) | 0.207 |
| | >10 yrs. | 18(58.1%) | 13(49.9%) | 0.475(0.196–1.152) | 0.099* | 0.611(0.169–2.202) | 0.451 |
| Previous first aid training | Yes | 58(56.9%) | 44(43.1%) | 3.149(1.810–5.478) | <0.001* | 2.902(1.612–5.227) | < 0.001** |
| | No | 25(20.5%) | 97(79.5%) | 1 | | 1 | |

** = significant at p≤ 0.05

* = associated at p≤ 0.25, COR-crude odes ration, AOR-adjusted odes ration.

towards first aid management of chocking [27]. The difference may be study method and availability of training facilities in Egypt.

This study showed that 45.5% of KG teachers have trained in first aid. It is lower than studies done in Iraq 73.5% [5]. South Africa, Cape town 74% [18] and Turk 73.6% [17] of the teachers received first-aid training. This is might be countries included first aid training in their educational curriculum as a course and mandatory to train.

Our study finding showed that teachers' previous first aid training experience had significantly associated with knowledge of chocking first aid management (AOR: 2.902, 95% CI: 1.612, 5.227, p<0.05). Kindergarten teachers with previous first aid training experience were about three times more knowledgeable than those who had not it. This is similar to a previous study done in Ethiopia where kindergarten teachers with the previous first aid training were three times more likely knowledgeable compared to those who had no training [19]. Similarly, studies conducted in China and Iran revealed significant association between previous first aid training and knowledge [15, 16].

Inadequate knowledge in the current study may be related with lack of first aid training given for KG teachers and not included it as a course in their curriculum.

Majority of respondents agreed that choking needs immediate management which was in line with the study done in Iraq [5].

The present study indicated that most of teachers' attitudes about first aid of choking were positive. This study is similar to a study done in Iraq where most of teachers were found to have a positive attitude toward first aid [5]. It is also comparable with a study conducted in Spain where more than 80% of the participants have agreed that everyone should have basic first aid knowledge [10]. The majority of study participants had poor practice, moreover a significant number of kindergarten teachers had bad and dangerous practices toward a choking victim like hitting at the back of the neck, putting fingers into the throat of a victim blindly, giving water to drink and blowing at the fontanel of the victim. The study finding is used as a baseline to intervene such malpractice for policymakers and responsible stakeholders for curriculum revision incorporating first aid training courses.

The limitation of this study are; One of the limitations is the study design being cross-sectional in that cause and effect association cannot be studied. Second, there may be a bias related with participants' level of understanding since the data collection tool was a self-administered questionnaire. Finally, lack of sufficient similar studies made limit comparison with other studies. Different mechanisms were tried to minimize bias. Some of these were performing pre-tests before actual data collection. Data collectors were explained to participants about the aim of the study, unclear ideas, and anything before and during data collection time.

## Conclusion

The knowledge and practice of kindergarten teachers towards provision of first aid for a choking child is low while their attitude is positive. The majority of teachers have encountered chocked children and provided first aid with a non-standard provision and some even had a life-threatening practice towards a choked victim. Having previous experience of first aid was significantly associated with knowledge of choking first aid provision. This needs urgent intervention to train teachers on first aid provision towards chocking management. It is recommended that first aid training is included in the kindergarten teachers' education curriculum as a course.

## Supporting information

**S1 File.**
(DOCX)

## Acknowledgments

We thank Addis Ababa University for providing us to conduct this study and its ethical approval. We also thank to administrators in the study schools who helped contact teachers for the survey. Lastly, Supervisor, Data collectors and study participants are to be thanked for their immense cooperation during data collection period.

## Author Contributions

**Conceptualization:** Ali Maalim Issack, Andualem Wubetie Aniley.

**Data curation:** Ali Maalim Issack, Andualem Wubetie Aniley.

**Formal analysis:** Ali Maalim Issack, Andualem Wubetie Aniley.

**Investigation:** Ali Maalim Issack.

**Methodology:** Ali Maalim Issack, Tilahun Jiru, Andualem Wubetie Aniley.

**Software:** Ali Maalim Issack, Tilahun Jiru, Andualem Wubetie Aniley.

**Supervision:** Ali Maalim Issack.

**Validation:** Tilahun Jiru, Andualem Wubetie Aniley.

**Visualization:** Ali Maalim Issack, Andualem Wubetie Aniley.

**Writing – original draft:** Ali Maalim Issack, Andualem Wubetie Aniley.

**Writing – review & editing:** Tilahun Jiru, Andualem Wubetie Aniley.

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
