## [Decision Letter · Decision Letter 0]

26 Apr 2021

PONE-D-20-32556

Assessment of knowledge, attitude and practice on first aid management of choking and associated factors among kindergarten teachers in Addis Ababa Governmental Schools, Addis Ababa, Ethiopia.     A cross-sectional institution-based study

PLOS ONE

Dear Dr. Wubetie,

Thank you for submitting your manuscript to PLOS ONE. After careful consideration, we feel that it has merit but does not fully meet PLOS ONE’s publication criteria as it currently stands. Therefore, we invite you to submit a revised version of the manuscript that addresses the points raised during the review process.

Please address all comments raised by the reviewers. However, please consider whether the references recommended by the reviewers should be cited or not.

We look forward to receiving your revised manuscript.

Sincerely,

Yann Benetreau, Ph.D.

Senior Editor, *PLOS ONE*

Journal Requirements:

2. Please include additional information regarding the survey or questionnaire used in the study and ensure that you have provided sufficient details that others could replicate the analyses. For instance, if you developed a questionnaire as part of this study and it is not under a copyright more restrictive than CC-BY, please include a copy, in both the original language and English, as Supporting Information.  If the original language is written in non-Latin characters, for example Amharic, Chinese, or Korean, please use a file format that ensures these characters are visible.

[We thank Addis Ababa University for supporting and providing funding for this study. Jigjiga University should be thanked for its unreserved support and sponsorship. Lastly, Supervisor, Data collectors and study participants are to be thanked for their immense cooperation during data collection period.]

 [The authors received no specific funding for this work.]

Reviewers' comments:

Reviewer's Responses to Questions

**Comments to the Author**

1. Is the manuscript technically sound, and do the data support the conclusions?

Reviewer #1: Partly

Reviewer #2: Yes

Reviewer #3: Yes

2. Has the statistical analysis been performed appropriately and rigorously? 

Reviewer #1: No

Reviewer #2: Yes

Reviewer #3: I Don't Know

3. Have the authors made all data underlying the findings in their manuscript fully available?

Reviewer #1: Yes

Reviewer #2: Yes

Reviewer #3: Yes

4. Is the manuscript presented in an intelligible fashion and written in standard English?

Reviewer #1: No

Reviewer #2: Yes

Reviewer #3: No

5. Review Comments to the Author

Reviewer #1: Good effort by the authors. However, there are some major concerns which are explained section wise as under

1. Introduction : Text is not supported with appropriate updated references .

There is a statement "As far as investigators’ knowledge, there is no study conducted in Ethiopia on chocking first aid". This study is part thesis submitted to Addis Ababa University, for partial fulfilment of the requirement for degree of masters in emergency medicine and critical care nursing and already available online .

2. Methods : Sampling technique multistage sampling is not clearly described

Sample size estimation is not correct

Data Collection tool : Development and validation of tool is not described adequately

Data collection procedure , inclusion & exclusion criteria is not clear

3. Results : How were the main outcomes of knowledge and practice adequacy defined? Please provide rationales for choosing the cut-off points for positive attitudes and adequate practice?

What was the response rate? How many school teachers were approached and out of them how many actually participated in study.

4. Discussion: Findings about factors associated with knowledge and skills of kindergarten teachers in the prevention and reduction of morbidity and mortality are not adequately compared with literature elsewhere, most of the study cited are from Ethiopia

5. Conclusion : The data presented in the manuscript must support the conclusions drawn. Conclusion mentioned in abstract and main manuscript are contradictory .

Implications of the findings, and what steps are needed to address the gaps are not addressed .The implications of this study on the generalizability of findings within Ethiopia are not addressed.

Please proofread and correct grammatical errors throughout the paper, there are lot of grammatical and spelling mistakes through out the manuscript. Need to revise to ensure that it is written in intelligible fashion and in standard English language .

Reviewer #2: First of all, I would like to share the need to carry out work like the one you present. They are necessary for the advancement of science in this sector. Life is the most important of all and even more so knowing how to intervene in the early stages of life as educators.

Thank you for allowing me to review this interesting study. Overall, the study has raised a very interesting point of discussion. I think this study has provided novel findings in this area in your country.

The specific term should be included according to the protocol established in the Basic Life Support (BLS) guidelines of the ERC (European Resuscitation Council) or AHA (American Heart Association) institutions.

The appropriate concept is OVACE (Foreign Body Airway Obstruction). It should be included since researchers and readers who work in the field of first aid is the term that refers to the objective of the proposed work. Review the reference institution in their continent and adapt the concept so that the international community knows what the work refers to.

Add the specific term to which the choking refers as a keyword.

It should be clarified that the training that kindergarten teachers have, since the authors reflect that 2 are graduates and 67.9% have a professional certificate. What is the training required to be able to practice in this age group in your country? It is understood that the study plans of these trainings do not teach first aid content, right?

I would like the authors to make a detailed explanation about the questionnaire to which the sample has been submitted. I would like to be able to analyze it, as well as describe the procedure to validate the instrument. It is true that they do it in the manuscript (in short), but I would like to know the procedure to follow in a more detailed way. As well as the statistical tests that have been carried out, both for the selection of the sample, and to pass it to 5% of the target population, to finally validate it and make it a valid and reliable instrument.

Reviewer #3: Authors present an interesting study about knowledge and attitudes towards choking management of kindergarten teachers. The importance of the topic is indubitable; schoolteachers have to be trained in choking management and first aid. However, the manuscript should be revised prior publication in an international journal. In first place, English must be revised, as well as all the mistakes and errors (some of them included in this review). In addition, the description of the results is redundant, duplicating data from the tables Please, see some comments and recommendations below:

Introduction:

l.61-62: Studies carried out in Iran, China and Turkey are mentioned. However, only two references (9 & 15) are placed into the text. Due to the limited literature existing about this topic, and the international readership of PLoS ONE, authors should try to include the majority of researches published in international journals. In this regard, there are studies published in other countries (i.e. Spain) in which schoolteachers (primary and pre-school) were asked for basic life support knowledge, including foreign body airway obstruction.

Methods:

It would be helpful to have the questionnaire. Please, add it as supplementary file. It cannot understand results reported in the tables without the main questionnaire. In addition, a better description of the questionnaire should be included.

l.116: “The variables were taken from previously studied literatures”. Which ones?

How was the questionnaire administered? How long did fill the questionnaire take?

Three sub-cities were randomly selected; but, why three?

Please, re-design Figure 1 to become more understandable.

l. 92-93: Please, replace “Data collection tool was adopted and modified from previously studied literatures (4),(13),(8) and (12)” with “A questionnaire from previous investigations were used (4,12,13,18)”.

Please, be consistent in the description of the results using both, absolute and relative frequencies, and only one decimal.

It is not clear how authors calculate the level of knowledge of KG teachers.

In expressions like (l.144) “Most respondents 211 (94.2%)…”, the absolute frequency cannot be included in the text in this way, it has to be in brackets or square brackets. Please, correct this in the whole manuscript.

Results:

Authors stated that the 37% of the KG teachers had adequate knowledge; based on what? It was not reported any data about what is adequate/inadequate knowledge.

Table 2: “Frecuency cy”. It should be corrected.

L.179-180: What does it mean that majority of the respondents have positive attitude towards providing first aid for a choking child? How did this data calculated?

Authors stated that 57.1% agreed that choking needs immediate management. Why did not you talk about those KG teaches who strongly agreed? This is an example that the description of the results in the texts is a duplication of the information reported in the tables. This has to be corrected in all the manuscript. Continuing with this example, if somebody strongly agree with something, at the same time agree; thus, although in the table the results are split in function of the questionnaire responses, in the text description it has to be used another way in order to avoid duplicate information.

Table 3: “Uncerta Disagree in N(%) N (%)”. It should be corrected.

Please, replace “Percent” with “Percentage” in all tables.

What do you mean with “Seventy-eight (34.8%) of the respondents did not know where to provide first aid”?

Table 5: Please, remove all data regarding no significant differences. “0.000” must not be used; it has to be replaced with “< 0.001”. CI lower and upper bounds should be separated by “-“. In addition, results of Table 5 should be revised; there are high differences between COR and AOR, is this correct? For example, in the case of Age (25-29), 0.156 becomes in 5.889.

Discussion:

Discussion should be start summarizing the main findings. Then, it should be follow the same order than results. The association between first aid training and knowledge was the last result reported, but it the first result discussed. Please, be consistent in this regard.

Again, the manuscript cited from China has no reference (l.237-238).

L.238-239: It does not understand the following sentence: “But a study finding in Egypt was higher than our finding where majority of teachers were knowledgeable towards first aid management of chocking”

Discussion should address how to achieve that KG teachers train in first aid matters as choking management; for example, including this contents in university degrees, which was recommended in previous publications. The following references should be to take into account to complement the introduction and the discussion:

https://doi.org/doi:10.1016/j.anpede.2018.10.013

https://doi.org/10.1097/EJA.0000000000001272

https://doi.org/10.1007/s00431-021-03971-x

https://doi.org/10.1016/j.resuscitation.2020.04.021

https://doi.org/10.1016/j.anpede.2019.10.005

6. PLOS authors have the option to publish the peer review history of their article (what does this mean?). If published, this will include your full peer review and any attached files.

Reviewer #1: No

Reviewer #2: No

Reviewer #3: No

---

## [Author Response · Author response to Decision Letter 0]

7 Jun 2021

To editor 

Thank you for giving me the opportunity to submit a revised draft of my manuscript titled - knowledge, attitude, and practice on first aid management of choking and associated factors among kindergarten teachers in Addis Ababa Governmental Schools, Addis Ababa, Ethiopia to PLOS ONE. I appreciate the time and effort that you and the reviewers have dedicated to providing your valuable feedback on my manuscript. I am grateful to the reviewers for their insightful comments on my paper. I have been able to incorporate changes to reflect most of the suggestions provided by the reviewers. I have highlighted in blue the changes within the manuscript. I hope it will now be suitable for publication in the PLOS ONE. 

With regards 

Andualem Wubetie, MSc 

Here is a point-by-point response to the reviewers’ comments and concerns. 

Comments from the editor

Comment 1: Please ensure that your manuscript meets PLOS ONE's style requirements, including those for file naming

Response: Thank you for your comment. We corrected it with PLOS ONE's style requirements. 

Comment 2: Please include additional information regarding the survey or questionnaire used in the study and ensure that you have provided sufficient details that others could replicate the analyses

Response: Thank you also for this comment. We have revised and corrected it. 

Comment 3: We note that you have provided funding information that is not currently declared in your Funding Statement. However, funding information should not appear in the Acknowledgments section or other areas of your manuscript.Please remove any funding-related text from the manuscript and let us know how you would like to update your Funding Statement

Response: Thank you for pointing these out. We agree with these comments. Therefore, we have corrected in the manuscript. 

Comments from Reviewer 

Reviewer 1: 

Comment 1: . Introduction :

1- Text is not supported with appropriate updated references.

Response: Thank you for pointing this out. We agree with this comment. Even if there is shortage of references, we have tried to correct. 

2: There is a statement "As far as investigators’ knowledge, there is no study conducted in Ethiopia on chocking first aid". This study is part thesis submitted to Addis Ababa University, for partial fulfilment of the requirement for degree of masters in emergency medicine and critical care nursing and already available online

Response: This is correct. This study is done for partial fulfilment of the requirement for degree of masters in emergency medicine and critical care nursing in Addis Ababa University and submitted to Addis Ababa university library which is a rule of the university for graduation requirement. We submitted this study to PLOS ONE for publication. 

 comment 2: Methods :

 Sampling technique multistage sampling is not clearly described

Sample size estimation is not correct

Data Collection tool : Development and validation of tool is not described adequately

Data collection procedure , inclusion & exclusion criteria is not clear

Response: Thank you also for pointing out this We have modified and tried to describe this point clearly. 

Comment 3: Results :

 How were the main outcomes of knowledge and practice adequacy defined? Please provide rationales for choosing the cut-off points for positive attitudes and adequate practice?

What was the response rate? How many school teachers were approached and out of them how many actually participated in study?

Response: Thank you for the comments. We have tried to correct it in the manuscript.

Comment 4: - Discussion: Findings about factors associated with knowledge and skills of kindergarten teachers in the prevention and reduction of morbidity and mortality are not adequately compared with literature elsewhere, most of the study cited are from Ethiopia

Response: We agree, we have modified and tried to describe pertinent findings of our study with previously done to emphasize this point. 

Comment 5: Conclusion : 

The data presented in the manuscript must support the conclusions drawn. Conclusion mentioned in abstract and main manuscript are contradictory .

Implications of the findings, and what steps are needed to address the gaps are not addressed .The implications of this study on the generalizability of findings within Ethiopia are not addressed.

Response: We agree with the points out. We have tried to correct your suggestion in the manuscript

Reviewer 2: 

Comment 1: The specific term should be included according to the protocol established in the Basic Life Support (BLS) guidelines of the ERC (European Resuscitation Council) or AHA (American Heart Association) institutions

Response: Thank you for pointing out this. We corrected it. 

Comment 2:The appropriate concept is OVACE (Foreign Body Airway Obstruction). It should be included since researchers and readers who work in the field of first aid is the term that refers to the objective of the proposed work. Review the reference institution in their continent and adapt the concept so that the international community knows what the work refers to.

Add the specific term to which the choking refers as a keyword

Response: We agreed and corrected it. 

Comment 3: It should be clarified that the training that kindergarten teachers have, since the authors reflect that 2 are graduates and 67.9% have a professional certificate. What is the training required to be able to practice in this age group in your country? It is understood that the study plans of these trainings do not teach first aid content, right?

Response: we revised and tried to clarify it in the manuscript. 

Comment 4: I would like the authors to make a detailed explanation about the questionnaire to which the sample has been submitted. I would like to be able to analyze it, as well as describe the procedure to validate the instrument. It is true that they do it in the manuscript (in short), but I would like to know the procedure to follow in a more detailed way. As well as the statistical tests that have been carried out, both for the selection of the sample, and to pass it to 5% of the target population, to finally validate it and make it a valid and reliable instrument.

Response: thank you for pointing out this. We have revised and tried to explain in detail.

Reviewer 3: 

Comment 1: Introduction:

l.61-62: Studies carried out in Iran, China and Turkey are mentioned. However, only two references (9 & 15) are placed into the text. Due to the limited literature existing about this topic, and the international readership of PLoS ONE, authors should try to include the majority of researches published in international journals. In this regard, there are studies published in other countries (i.e. Spain) in which schoolteachers (primary and pre-school) were asked for basic life support knowledge, including foreign body airway obstruction.

Response: Thank you for your comment. We have added and tried to correct in the manuscript 

Comment 2: Methods:

It would be helpful to have the questionnaire. Please, add it as supplementary file. It cannot understand results reported in the tables without the main questionnaire. In addition, a better description of the questionnaire should be included.

l.116: “The variables were taken from previously studied literatures”. Which ones?

How was the questionnaire administered? How long did fill the questionnaire take?

Three sub-cities were randomly selected; but, why three?

Please, re-design Figure 1 to become more understandable.

l. 92-93: Please, replace “Data collection tool was adopted and modified from previously studied literatures (4),(13),(8) and (12)” with “A questionnaire from previous investigations were used (4,12,13,18)”.

Please, be consistent in the description of the results using both, absolute and relative frequencies, and only one decimal.

It is not clear how authors calculate the level of knowledge of KG teachers.

In expressions like (l.144) “Most respondents 211 (94.2%)…”, the absolute frequency cannot be included in the text in this way, it has to be in brackets or square brackets. Please, correct this in the whole manuscript.

Response: Thank you for pointing out these constructive comments. We agreed on comments and corrected accordingly. 

Comment 3: Results:

Comment 3.1 Authors stated that the 37% of the KG teachers had adequate knowledge; based on what?

Response: It is the total participants’ who scored mean and above mean of knowledge questions.

Comment 3.2: It was not reported any data about what is adequate/inadequate knowledge.

Response: we reported in table form for all knowledge questions how much they answered or knowledgeable and also tried to explain major findings in sentence. 

Comment 3.3: L.179-180: What does it mean that majority of the respondents have positive attitude towards providing first aid for a choking child? How did this data calculated?

Response: respondents scored above mean of attitude questions or agree and strongly agree respondents were considered to have positive attitude. 

Comment 3.4: Authors stated that 57.1% agreed that choking needs immediate management. Why did not you talk about those KG teaches who strongly agreed? This is an example that the description of the results in the texts is a duplication of the information reported in the tables. This has to be corrected in all the manuscript. Continuing with this example, if somebody strongly agree with something, at the same time agree; thus, although in the table the results are split in function of the questionnaire responses, in the text description it has to be used another way in order to avoid duplicate information.

Response: Thank you for the constructive comment. We corrected it throughout the manuscript. 

Table 3: “Uncerta Disagree in N(%) N (%)”. It should be corrected.

Please, replace “Percent” with “Percentage” in all tables.

What do you mean with “Seventy-eight (34.8%) of the respondents did not know where to provide first aid”?

Table 5: Please, remove all data regarding no significant differences. “0.000” must not be used; it has to be replaced with “< 0.001”. CI lower and upper bounds should be separated by “-“. In addition, results of Table 5 should be revised; there are high differences between COR and AOR, is this correct? For example, in the case of Age (25-29), 0.156 becomes in 5.889.

Response: Thank you also for pointing out these points. We have modified and tried to describe these in the manuscript. 

Comment 4: Discussion:

Discussion should be start summarizing the main findings. Then, it should be follow the same order than results. The association between first aid training and knowledge was the last result reported, but it the first result discussed. Please, be consistent in this regard.

Again, the manuscript cited from China has no reference (l.237-238).

L.238-239: It does not understand the following sentence: “But a study finding in Egypt was higher than our finding where majority of teachers were knowledgeable towards first aid management of chocking”

Discussion should address how to achieve that KG teachers train in first aid matters as choking management; for example, including this contents in university degrees, which was recommended in previous publications

Response: We agreed with the comments and tried to correct it accordingly.

Additional clarifications 

In addition to the above comments, spelling and grammatical errors pointed out by the reviewers have been tried to correct. 

We look forward to hear from you in due time regarding our submission and to respond to any further questions and comments you may have. 

Sincerely, 

Andualem Wubetie Aniley

---

## [Editor Report · Decision Letter 1]

23 Jun 2021

PONE-D-20-32556R1

Assessment of knowledge, attitude and practice on first aid management of choking and associated factors among kindergarten teachers in Addis Ababa Governmental Schools, Addis Ababa, Ethiopia.     A cross-sectional institution-based study

PLOS ONE

Dear Dr. Andualem Wubetie, Msc

Thank you for submitting your manuscript to PLOS ONE. After careful consideration, we feel that it has merit but does not fully meet PLOS ONE’s publication criteria as it currently stands. Therefore, we invite you to submit a revised version of the manuscript that addresses the points raised during the review process.

We look forward to receiving your revised manuscript.

Kind regards,

Sergio García López, Ph.D.

Academic Editor

PLOS ONE

Journal Requirements:

Additional Editor Comments (if provided):

Dear author,

First of all, I have to thank you for the effort you have made so far with your manuscript.

I think it is a work that will be a reference in your country on the subject of study.

However, there are some considerations that the reviewers have made that you have not addressed. I invite you to be able to review and correct them. In addition, those that are not carried out, the justified reason must be detailed in order to be evaluated by the editorial committee.

Likewise, they have been emphasized in the incorporation by a reviewer of some references that are considered essential for their work. Keep in mind that there are not many scientific studies in this matter, and the existing ones should be reflected, since this facilitates the advancement and contrast of the results and consequently, scientific knowledge advances.

I invite you to review these questions and make small corrections in order to be successful in this last phase of the process.

---

## [Author Response · Author response to Decision Letter 1]

27 Jun 2021

To editors, 

Thank you for giving me the opportunity to submit a revised draft of my manuscript titled - knowledge, attitude, and practice on first aid management of choking and associated factors among kindergarten teachers in Addis Ababa Governmental Schools, Addis Ababa, Ethiopia to PLOS ONE. I appreciate the time and effort that you and the reviewers have dedicated to providing your valuable feedback on my manuscript. I am grateful to the reviewers for their insightful comments on my paper. I have been able to incorporate changes to reflect most of the suggestions provided by the reviewers. I have highlighted in blue the changes within the manuscript. I hope it will now be suitable for publication in the PLOS ONE. 

With regards 

Andualem Wubetie, MSc 

Here is a point-by-point response to the reviewers’ comments and concerns. 

Comments from the editor

Comment 1: Please ensure that your manuscript meets PLOS ONE's style requirements, including those for file naming

Response: Thank you for your comment. We corrected it with PLOS ONE's style requirements. 

Comment 2: Please include additional information regarding the survey or questionnaire used in the study and ensure that you have provided sufficient details that others could replicate the analyses

Response: Thank you also for this comment. We have revised and corrected it. 

Comment 3: We note that you have provided funding information that is not currently declared in your Funding Statement. However, funding information should not appear in the Acknowledgments section or other areas of your manuscript.Please remove any funding-related text from the manuscript and let us know how you would like to update your Funding Statement

Response: Thank you for pointing these out. We agree with these comments. Therefore, we have corrected in the manuscript. 

Additional comments from editors

Comment 1: Journal Requirements: Please review your reference list to ensure that it is complete and correct. If you have cited papers that have been retracted, please include the rationale for doing so in the manuscript text, or remove these references and replace them with relevant current references. Any changes to the reference list should be mentioned in the rebuttal letter that accompanies your revised manuscript. If you need to cite a retracted article, indicate the article’s retracted status in the References list and also include a citation and full reference for the retraction notice.

Response: Thank you for adding these essential comments. We have revised references based on the referencing style for its completeness and correct order using updated citation manager. In addition, we have incorporated relevant current references and to cite suggestive evidences based on comments raised from reviewers. Additional references are (2, 5-8,10-12,22,23,26)

2. REPUBLIC OF GUYANA, Ministry of Public Health. Standard Treatment Guidelines for Primary Health Care. second edition. Guyana; 2015. 4–5 p.

5. Ibrahim H. Mustafa SSH. KNOWLEDGE AND ATTITUDE OF PRIMARY SCHOOL TEACHERS REGARDING CHOKING’S FIRST AID IN ERBIL CITY- KURDISTAN REGION - IRAQ. 2016 Oct;8(2):37–9. 

6. Narayanan T. MN Med C, Med PF. Awareness , attitudes and practices of first aid among school teachers in Mangalore south India. 2015;7(4):274-81. 

7. Georgiou M, Koenraad G Monsieurs, Nikolaos Nikolaou. KIDS SAVE LIVES: ERC Position statement on schoolteachers’ education and qualification in resuscitation. ELSEVIER,EUROPEAN RESUSCITATION COUNCIL. 2020 Oct 25;2020:8 7 _9 0. 

8. Cristian Abelairas-Gómez, Aida Carballo-Fazanes, Santiago Martínez-Isasid,Sergio López-García, Javier Rico-Díazb, Antonio Rodríguez-Núnez. Knowledge and attitudes on first aid and basic lifesupport of pre- and elementary school teachers andparents. analesdepeditria. 2019 Oct 27;1–6. /. 

10. The British Red Cross Society British Red Cross First Aid Resources. [2020-05-20]. https://www.redcross.org.uk/first-aid/. 

11. American Heart Association AHA information. 2020. May 25, [2020-05-25]. https://www.heart.org/. 

12. Robert Greif, Lockey A, Jan Breckwoldt, Patricia Conaghan. European Resuscitation Council Guidelines 2021: Education for resuscitation. ELSEVIER, R E S U S C I T A T I O N. 2 0 2 1;161(2021):3 8 8-4 0 7.

22. Steve Bennett," lony Woods;, Winith M. Iiyanage & Duane . Smithd. SIMPLIFIED GENERAL METHOD FOR (CLUSTER-SANllabE SURVEYS OF WEALTH IN DEVELOPING GObfNTIFSfilES. 1991;44. 

23. Public Health Regional, Surveillance Team,Steven Ramsey. A Guide to Sampling for Community Health Assessments and Other Projects. N C Cent Public Health Prep. :1–5. 

26. Shaima Shaban Mohamad, Dr. Afkar Ragab Mohamad. First Aid Program For Nursery School Teachers,Egypt. 2018 Aug;7(4):01–9.

Comment 2: There are some considerations that the reviewers have made that you have not addressed. I invite you to be able to review and correct them. In addition, those that are not carried out, the justified reason must be detailed in order to be evaluated by the editorial committee.

Response: Thank you for pointing out these that we did not addressed. We have corrected comments in the manuscript but not incorporated in the response letter assumed to be seen in the manuscript. Now, we have done it for every comment. 

Comment 3: Reviewers have been emphasized in the incorporation by a reviewer of some references that are considered essential for their work. Keep in mind that there are not many scientific studies in this matter, and the existing ones should be reflected, since this facilitates the advancement and contrast of the results and consequently, scientific knowledge advances.

Response: We have incorporated essential current references in the manuscript. 

Comments from Reviewer 

Reviewer 1: 

Comment 1: . Introduction :

1- Text is not supported with appropriate updated references.

Response: Thank you for pointing this out. We agree with this comment. Even if there is shortage of references, we have tried to correct. 

2: There is a statement "As far as investigators’ knowledge, there is no study conducted in Ethiopia on chocking first aid". This study is part thesis submitted to Addis Ababa University, for partial fulfilment of the requirement for degree of masters in emergency medicine and critical care nursing and already available online

Response: This is correct. This study is done for partial fulfilment of the requirement for degree of masters in emergency medicine and critical care nursing in Addis Ababa University and submitted to Addis Ababa university library which is a rule of the university for graduation requirement. We submitted this study to PLOS ONE for publication. 

 comment 2: Methods :

1.Sampling technique multistage sampling is not clearly described

Response: Thank you. We have tried to describe clearly with evidence used sampling technique. 

2. Sample size estimation is not correct

Response: Thank you for insight comment for sample size estimation. We have explained it how sample was calculated and estimated in the manuscript. 

3. Data Collection tool : Development and validation of tool is not described adequately

 Response: Thank you for advising us to describe adequately about it. We have tried to describe adequately about data collection tool and its development and validation with evidenced citation in the manuscript. 

4. Data collection procedure , inclusion & exclusion criteria is not clear

5. Response: Thank you also for pointing out this. We have modified and tried to describe this point clearly in the manuscript. 

Comment 3: Results :

1. How were the main outcomes of knowledge and practice adequacy defined? Please provide rationales for choosing the cut-off points for positive attitudes and adequate practice?

Response: Thank you. We have added operational definition for these terms as:

Adequate knowledge – a participant who scored a mean and above for knowledge questions.

Inadequate knowledge – a participant who scored below mean for knowledge questions.

Positive Attitude- a participant who answered agree and strongly for attitude questions.

Negative attitude – a participant who answered disagree and strongly disagree for attitude questions.

Good Practice– a participant who scored 80%and above of practical questions (according to AHA pediatric basic life support and advanced life support 2020)

Poor practice – a participant who scored below 80%of practical questions(AHA,PBLS and PALS 2020)

What was the response rate? How many school teachers were approached and out of them how many participated in study?

Response: Thank you for the comments. Out of 235 participants, 224 were correctly responded the provided questions with a response rate of 95%. We have tried to correct it in the manuscript. (p-8-line 200-202)

Comment 4: - Discussion: Findings about factors associated with knowledge and skills of kindergarten teachers in the prevention and reduction of morbidity and mortality are not adequately compared with literature elsewhere, most of the study cited are from Ethiopia

Response: Thank you for suggesting this comment., We have modified and tried to describe pertinent findings of our study with previously done to emphasize this point.(p-15-16) 

Comment 5: Conclusion : 

The data presented in the manuscript must support the conclusions drawn. Conclusion mentioned in abstract and main manuscript are contradictory .

Implications of the findings, and what steps are needed to address the gaps are not addressed .The implications of this study on the generalizability of findings within Ethiopia are not addressed.

Response: We agree with the points out. We have tried to correct your suggestion in the manuscript

Reviewer 2: 

Comment 1: The specific term should be included according to the protocol established in the Basic Life Support (BLS) guidelines of the ERC (European Resuscitation Council) or AHA (American Heart Association) institutions

Response: Thank you for pointing out this. We incorporated it in our manuscript with updated references(p 3-4, line 70 -75) 

Comment 2:The appropriate concept is OVACE (Foreign Body Airway Obstruction). It should be included since researchers and readers who work in the field of first aid is the term that refers to the objective of the proposed work. Review the reference institution in their continent and adapt the concept so that the international community knows what the work refers to.

Response: Thank you for your advice. We have incorporated these points in our manuscript(p-3, line 46-50) 

Comment 3: Add the specific term to which the choking refers as a keyword

Response: we added it as key word.

Comment 4: It should be clarified that the training that kindergarten teachers have, since the authors reflect that 2 are graduates and 67.9% have a professional certificate. What is the training required to be able to practice in this age group in your country? It is understood that the study plans of these trainings do not teach first aid content, right?

Response: Thank you for your insight comment and advice. certificate level(secondary school complete with additional two years teachers education training); which is a minimum requirement to be KG teacher in Ethiopia and 2 (0.9%) were Bachelor Degree holders. We revised and tried to clarify it in the manuscript ( p-8,line 203-205). There is no first aid training in their curriculum. So, we recommended the responsible bodies to incorporate it. But we assessed educational level if it might be a factor for knowledge, attitude or practice for chocking first aid. 

Comment 5: I would like the authors to make a detailed explanation about the questionnaire to which the sample has been submitted. I would like to be able to analyze it, as well as describe the procedure to validate the instrument. It is true that they do it in the manuscript (in short), but I would like to know the procedure to follow in a more detailed way. As well as the statistical tests that have been carried out, both for the selection of the sample, and to pass it to 5% of the target population, to finally validate it and make it a valid and reliable instrument.

Response: thank you for pointing out this. We have revised and tried to explain in detail in the manuscript with track changed highlighted in blue (p, 5-6) 

Reviewer 3: 

Comments : Introduction:

comment 1: . l.61-62: Studies carried out in Iran, China and Turkey are mentioned. However, only two references (9 & 15) are placed into the text.

Response: Thank you. We corrected it (p-4, line 76-77)

Comment 2: Due to the limi;ted literature existing about this topic, and the international readership of PLoS ONE, authors should try to include the majority of researches published in international journals. In this regard, there are studies published in other countries (i.e. Spain) in which schoolteachers (primary and pre-school) were asked for basic life support knowledge, including foreign body airway obstruction.

Response: Thank you for your suggestion to add these studies. We have incorporated in the manuscript introduction and discussion part. 

Methods:

Comment 1: It would be helpful to have the questionnaire. Please, add it as supplementary file. It cannot understand results reported in the tables without the main questionnaire. In addition, a better description of the questionnaire should be included.

Response: Thank you for pointing out it. We have attached the questionnaire as supplementary file both English and local language (Amharic) version. We used Amharic version to collect the data. We have described clearly in the manuscript. (p,5-6, line 118-138) 

comment 2: l.116: “The variables were taken from previously studied literatures”. Which ones?

How was the questionnaire administered? How long did fill the questionnaire take?

Response: thank you: we have clearly described it in the manuscript of methodology part( p,5-6). Average time taken to fill questionnaire was 10 minute(p,6, line 150) 

Three sub-cities were randomly selected; but, why three?

comment 3: Please, re-design Figure 1 to become more understandable.

Response: we have redesigned with more clear form (see attached figure 1) 

Comment 4: l. 92-93: Please, replace “Data collection tool was adopted and modified from previously studied literatures (4),(13),(8) and (12)” with “A questionnaire from previous investigations were used (4,12,13,18)”.

Response: We corrected it (p,5, line 121). 

comment 5: Please, be consistent in the description of the results using both, absolute and relative frequencies, and only one decimal.

Response: Thank you. we corrected based on comments. 

comment 6: It is not clear how authors calculate the level of knowledge of KG teachers.

Response: Thank you. We have used the mean score of knowledge questions to calculate participants’ level of knowledge. To say knowledgeable, we used a score of mean and above of knowledge questions. Please see descriptions more in the manuscript with track change highlighted in blue (p,5. Line 123-128). 

comment 7: In expressions like (l.144) “Most respondents 211 (94.2%)…”, the absolute frequency cannot be included in the text in this way, it has to be in brackets or square brackets. Please, correct this in the whole manuscript.

Response: Thank you for pointing out comments. We have written the manuscript based on the previously published article style and corrected accordingly. 

Results:

Comment 1: Authors stated that the 37% of the KG teachers had adequate knowledge; based on what?

Response: It is the total number of participants’ who scored mean and above mean of knowledge questions.

Comment 2: It was not reported any data about what is adequate/inadequate knowledge.

Response: we reported in table form for all knowledge questions how much they answered or knowledgeable and tried to explain major findings in sentence as knowledgeable and not (p,9-10) 

Cooment 3: Table 2: “Frecuency cy”. It should be corrected

Response: Thank you. We corrected it. (p,10. Line 234-235)

Comment 4: L.179-180: What does it mean that majority of the respondents have positive attitude towards providing first aid for a choking child? How did this data calculated?

Response: Respondents answered attitude questions as agree and strongly agree were considered to have positive attitude. Attitude questions were dichotomized as positive attitude and negative attitude. The score mean and above attitude question was considered as positive attitude and below mean was negative attitude. 

Comment 5: Authors stated that 57.1% agreed that choking needs immediate management. Why did not you talk about those KG teaches who strongly agreed? This is an example that the description of the results in the texts is a duplication of the information reported in the tables. This has to be corrected in all the manuscript. Continuing with this example, if somebody strongly agree with something, at the same time agree; thus, although in the table the results are split in function of the questionnaire responses, in the text description it has to be used another way in order to avoid duplicate information.

Response: Thank you for the constructive comment. We corrected it throughout the manuscript. 

Comment 6: Table 3: “Uncerta Disagree in N(%) N (%)”. It should be corrected.

Response: Thank you. We have corrected (p,11. Line 250-251)

Comment 7: Please, replace “Percent” with “Percentage” in all tables.

Response : Thank you. We have corrected it. 

Comment 8: What do you mean with “Seventy-eight (34.8%) of the respondents did not know where to provide first aid”?

Response: it is to mean the site of our body to provide first aid (Heimlich man over or back slap) for complete airway obstruction. (p,15. Line 260-261). 

Comment 9: Table 5: Please, remove all data regarding no significant differences. “0.000” must not be used; it has to be replaced with “< 0.001”. CI lower and upper bounds should be separated by “-“. In addition, results of Table 5 should be revised; there are high differences between COR and AOR, is this correct? For example, in the case of Age (25-29), 0.156 becomes in 5.889.

Response: Thank you for your insight comment and advise. We have modified and tried to describe these in the manuscript (p,14. Table 5) 

Discussion:

Comment 1: Discussion should be start summarizing the main findings. Then, it should be follow the same order than results. The association between first aid training and knowledge was the last result reported, but it the first result discussed. Please, be consistent in this regard.

Response: Thank you for your guidance in comment. We have tried to correct based on comments accordingly. 

Comment 2: Again, the manuscript cited from China has no reference (l.237-238).

Response: We corrected it (p,15. Line 316)

Comment 3: L.238-239: It does not understand the following sentence: “But a study finding in Egypt was higher than our finding where majority of teachers were knowledgeable towards first aid management of chocking”

Response: It is to mean that teachers in Egypt were more knowledgeable compared with this finding. (p,15. Line 304-306). 

Comment 4: Discussion should address how to achieve that KG teachers train in first aid matters as choking management; for example, including this contents in university degrees, which was recommended in previous publications

Response: Thank you your comment and advice. We incorporate it in the manuscript in the discussion and recommendation part as first aid training to be included in the teachers’ educational curriculum.(p,16. Line 344-346) 

Comment 5: The following references should be to take into account to complement the introduction and the discussion:

https://doi.org/doi:10.1016/j.anpede.2018.10.013

https://doi.org/10.1097/EJA.0000000000001272

https://doi.org/10.1007/s00431-021-03971-x

https://doi.org/10.1016/j.resuscitation.2020.04.021

https://doi.org/10.1016/j.anpede.2019.10.005

Response: Thank you for suggesting us to use references. We have used these articles both in the introduction and discussion part in the manuscript. We greatly appreciate all reviewers’ thoughtful review and recommendations in the manuscript. 

While revising your submission, please upload your figure files to the Preflight Analysis and Conversion Engine (PACE) digital diagnostic tool, https://pacev2.apexcovantage.com/. PACE helps ensure that figures meet PLOS requirements. To use PACE, you must first register as a user. Registration is free. Then, login and navigate to the UPLOAD tab, where you will find detailed instructions on how to use the tool. If you encounter any issues or have any questions when using PACE, please email PLOS at figures@plos.org. Please note that Supporting Information files do not need this step

Response: We have registered with PACE and the figures meets PLOS requirements. We downloaded from the PACE and uploaded as Figure 1. 

Additional clarifications 

In addition to the above comments, spelling and grammatical errors pointed out by the reviewers have been tried to correct. 

We look forward to hear from you in due time regarding our submission and to respond to any further questions and comments you may have. 

Sincerely, 

Andualem Wubetie Aniley _____________

27/06/2021.

---

## [Editor Report · Decision Letter 2]

7 Jul 2021

PONE-D-20-32556R2

Assessment of knowledge, attitude and practice on first aid management of choking and associated factors among kindergarten teachers in Addis Ababa Governmental Schools, Addis Ababa, Ethiopia.     A cross-sectional institution-based study

PLOS ONE

Dear Dr. Mr Andualem Wubetie

Thank you for submitting your manuscript to PLOS ONE. After careful consideration, we feel that it has merit but does not fully meet PLOS ONE’s publication criteria as it currently stands. Therefore, we invite you to submit a revised version of the manuscript that addresses the points raised during the review process.

We look forward to receiving your revised manuscript.

Kind regards,

Sergio García López, Ph.D.

Academic Editor

PLOS ONE

Journal Requirements:

Additional Editor Comments (if provided):

I thank the authors for the effort they have made with the changes to the manuscript.

They have improved the previous version. However, the considerations of a reviewer remain unaddressed. They have not incorporated the quotes that you have delicately and professionally suggested. In an area where there is little scientific evidence, attention must be paid to the work carried out that is in line with the one presented. It is essential that the authors incorporate them. If not, the reason for not joining must be justified. They should understand that reviewers carefully try to improve their manuscript for international reference. This gives your work a higher quality and gives the magazine added value. Based on the above, I suggest that you be careful and incorporate the quotes

---

## [Author Response · Author response to Decision Letter 2]

11 Jul 2021

To Editors

 Thank you again for giving me the opportunity to submit a revised draft of my manuscript titled - knowledge, attitude, and practice on first aid management of choking and associated factors among kindergarten teachers in Addis Ababa Governmental Schools, Addis Ababa, Ethiopia to PLOS ONE. I appreciate the time and effort that you and the reviewers have dedicated to providing your valuable feedback on my manuscript. I am grateful to the reviewers for their insightful comments on my paper. I have been able to incorporate changes to reflect most of the suggestions provided by the reviewers. I have highlighted in blue the changes within the manuscript. I hope it will now be suitable for publication in the PLOS ONE. 

With regards 

Andualem Wubetie, MSc 

Here is a point-by-point response to the reviewers’ comments and concerns. 

Comments from the editor

PLOS ONE

Journal Requirements:

Response: Thank you for your insight and constructive comment to correct these comments. We have reviewed all references and corrected accordingly. 

We corrected papers that have been listed in the reference but incorrectly cited in the manuscript. Correctly cited papers are reference 15,16,17. (p-4, line 83)

We have removed papers that are not recent and considering as not much relevant and replaced with current papers according to your suggested comments and reviewers’

Removed papers are. 

(American Academy of Pediatrics, Committee on School Health. Guidelines for emergency medical care in school. Pediatrics. 2001;107:435– 436. 

Altayehu M. Assessment of General approach for appropriate pediatric first aid and associated factors among primary school teachers in Addis Ababa,Ethiopia; 2015. 2015;1–55.

Meral Bayat MB. Evaluating First-aid Knowledge and Attitudes of a Sample of Turkish Primary School Teachers. 2007 Oct 1;33(5):428–32.)

Additional comments 

Additional Editor Comments (if provided):

I thank the authors for the effort they have made with the changes to the manuscript.

They have improved the previous version. However, the considerations of a reviewer remain unaddressed. They have not incorporated the quotes that you have delicately and professionally suggested. In an area where there is little scientific evidence, attention must be paid to the work carried out that is in line with the one presented. It is essential that the authors incorporate them. If not, the reason for not joining must be justified. They should understand that reviewers carefully try to improve their manuscript for international reference. This gives your work a higher quality and gives the magazine added value. Based on the above, I suggest that you be careful and incorporate the quotes

Response: 

Thank you also for the comments. We tried to incorporate the comments in the manuscript. 

To Reviewers 

Reviewer 1: 

Comment 1: . Introduction :

1- Text is not supported with appropriate updated references.

Response: Thank you for pointing this out. We agree with this comment. Even if there is shortage of references, we have tried to correct. 

2: There is a statement "As far as investigators’ knowledge, there is no study conducted in Ethiopia on chocking first aid". This study is part thesis submitted to Addis Ababa University, for partial fulfilment of the requirement for degree of masters in emergency medicine and critical care nursing and already available online

Response: This is correct. This study is done for partial fulfilment of the requirement for degree of masters in emergency medicine and critical care nursing in Addis Ababa University and submitted to Addis Ababa university library which is a rule of the university for graduation requirement. We submitted this study to PLOS ONE for publication. 

 comment 2: Methods :

1.Sampling technique multistage sampling is not clearly described

Response: Thank you. We have tried to describe clearly with evidence used sampling technique. 

2. Sample size estimation is not correct

Response: Thank you for insight comment for sample size estimation. We have explained it how sample was calculated and estimated in the manuscript. 

3. Data Collection tool : Development and validation of tool is not described adequately

 Response: Thank you for advising us to describe adequately about it. We have tried to describe adequately about data collection tool and its development and validation with evidenced citation in the manuscript. 

4. Data collection procedure , inclusion & exclusion criteria is not clear

5. Response: Thank you also for pointing out this. We have modified and tried to describe this point clearly in the manuscript. 

Comment 3: Results :

1. How were the main outcomes of knowledge and practice adequacy defined? Please provide rationales for choosing the cut-off points for positive attitudes and adequate practice?

Response: Thank you. We have added operational definition for these terms as:

Adequate knowledge – a participant who scored a mean and above for knowledge questions.

Inadequate knowledge – a participant who scored below mean for knowledge questions.

Positive Attitude- a participant who answered agree and strongly for attitude questions.

Negative attitude – a participant who answered disagree and strongly disagree for attitude questions.

Good Practice– a participant who scored 80%and above of practical questions (according to AHA pediatric basic life support and advanced life support 2020)

Poor practice – a participant who scored below 80%of practical questions(AHA,PBLS and PALS 2020)

What was the response rate? How many school teachers were approached and out of them how many participated in study?

Response: Thank you for the comments. Out of 235 participants, 224 were correctly responded the provided questions with a response rate of 95%. We have tried to correct it in the manuscript. (p-8-line 209-216))

Comment 4: - Discussion: Findings about factors associated with knowledge and skills of kindergarten teachers in the prevention and reduction of morbidity and mortality are not adequately compared with literature elsewhere, most of the study cited are from Ethiopia

Response: Thank you for suggesting this comment., We have modified and tried to describe pertinent findings of our study with previously done to emphasize this point.(p-15-16) 

Comment 5: Conclusion : 

The data presented in the manuscript must support the conclusions drawn. Conclusion mentioned in abstract and main manuscript are contradictory .

Implications of the findings, and what steps are needed to address the gaps are not addressed .The implications of this study on the generalizability of findings within Ethiopia are not addressed.

Response: We agree with the points out. We have tried to correct your suggestion in the manuscript

Reviewer 2: 

Comment 1: The specific term should be included according to the protocol established in the Basic Life Support (BLS) guidelines of the ERC (European Resuscitation Council) or AHA (American Heart Association) institutions

Response: Thank you for pointing out this. We incorporated it in our manuscript with updated references(p 4, line 76 -80) 

Comment 2:The appropriate concept is OVACE (Foreign Body Airway Obstruction). It should be included since researchers and readers who work in the field of first aid is the term that refers to the objective of the proposed work. Review the reference institution in their continent and adapt the concept so that the international community knows what the work refers to.

Response: Thank you for your advice. We have incorporated these points in our manuscript(p-3, line 46-50) 

Comment 3: Add the specific term to which the choking refers as a keyword

Response: we added it as key word.

Comment 4: It should be clarified that the training that kindergarten teachers have, since the authors reflect that 2 are graduates and 67.9% have a professional certificate. What is the training required to be able to practice in this age group in your country? It is understood that the study plans of these trainings do not teach first aid content, right?

Response: Thank you for your insight comment and advice. certificate level(secondary school complete with additional two years teachers education training); which is a minimum requirement to be KG teacher in Ethiopia and 2 (0.9%) were Bachelor Degree holders. We revised and tried to clarify it in the manuscript ( p-8,line 203-205). There is no first aid training in their curriculum. So, we recommended the responsible bodies to incorporate it. But we assessed educational level if it might be a factor for knowledge, attitude or practice for chocking first aid. 

Comment 5: I would like the authors to make a detailed explanation about the questionnaire to which the sample has been submitted. I would like to be able to analyze it, as well as describe the procedure to validate the instrument. It is true that they do it in the manuscript (in short), but I would like to know the procedure to follow in a more detailed way. As well as the statistical tests that have been carried out, both for the selection of the sample, and to pass it to 5% of the target population, to finally validate it and make it a valid and reliable instrument.

Response: thank you for pointing out this. We have revised and tried to explain in detail in the manuscript with track changed highlighted in blue (p, 5-6) 

Reviewer 3: 

Comments : Introduction:

comment 1: . l.61-62: Studies carried out in Iran, China and Turkey are mentioned. However, only two references (9 & 15) are placed into the text.

Response: Thank you. We corrected it (p-4, line 76-77)

Comment 2: Due to the limi;ted literature existing about this topic, and the international readership of PLoS ONE, authors should try to include the majority of researches published in international journals. In this regard, there are studies published in other countries (i.e. Spain) in which schoolteachers (primary and pre-school) were asked for basic life support knowledge, including foreign body airway obstruction.

Response: Thank you for your suggestion to add these studies. We have incorporated in the manuscript introduction and discussion part. 

Methods:

Comment 1: It would be helpful to have the questionnaire. Please, add it as supplementary file. It cannot understand results reported in the tables without the main questionnaire. In addition, a better description of the questionnaire should be included.

Response: Thank you for pointing out it. We have attached the questionnaire as supplementary file both English and local language (Amharic) version. We used Amharic version to collect the data. We have described clearly in the manuscript. (p,5-6, line 118-138) 

comment 2: l.116: “The variables were taken from previously studied literatures”. Which ones?

How was the questionnaire administered? How long did fill the questionnaire take?

Response: thank you: we have clearly described it in the manuscript of methodology part( p,5-6). Average time taken to fill questionnaire was 10 minute(p,6, line 150) 

Three sub-cities were randomly selected; but, why three?

comment 3: Please, re-design Figure 1 to become more understandable.

Response: we have redesigned with more clear form (see attached figure 1) 

Comment 4: l. 92-93: Please, replace “Data collection tool was adopted and modified from previously studied literatures (4),(13),(8) and (12)” with “A questionnaire from previous investigations were used (4,12,13,18)”.

Response: We corrected it (p,5, line 121). 

comment 5: Please, be consistent in the description of the results using both, absolute and relative frequencies, and only one decimal.

Response: Thank you. we corrected based on comments. 

comment 6: It is not clear how authors calculate the level of knowledge of KG teachers.

Response: Thank you. We have used the mean score of knowledge questions to calculate participants’ level of knowledge. To say knowledgeable, we used a score of mean and above of knowledge questions. Please see descriptions more in the manuscript with track change highlighted in blue (p,5. Line 123-128). 

comment 7: In expressions like (l.144) “Most respondents 211 (94.2%)…”, the absolute frequency cannot be included in the text in this way, it has to be in brackets or square brackets. Please, correct this in the whole manuscript.

Response: Thank you for pointing out comments. We have written the manuscript based on the previously published article style and corrected accordingly. 

Results:

Comment 1: Authors stated that the 37% of the KG teachers had adequate knowledge; based on what?

Response: It is the total number of participants’ who scored mean and above mean of knowledge questions.

Comment 2: It was not reported any data about what is adequate/inadequate knowledge.

Response: we reported in table form for all knowledge questions how much they answered or knowledgeable and tried to explain major findings in sentence as knowledgeable and not (p,9-10) 

Cooment 3: Table 2: “Frecuency cy”. It should be corrected

Response: Thank you. We corrected it. (p,10. Line 234-235)

Comment 4: L.179-180: What does it mean that majority of the respondents have positive attitude towards providing first aid for a choking child? How did this data calculated?

Response: Respondents answered attitude questions as agree and strongly agree were considered to have positive attitude. Attitude questions were dichotomized as positive attitude and negative attitude. The score mean and above attitude question was considered as positive attitude and below mean was negative attitude. 

Comment 5: Authors stated that 57.1% agreed that choking needs immediate management. Why did not you talk about those KG teaches who strongly agreed? This is an example that the description of the results in the texts is a duplication of the information reported in the tables. This has to be corrected in all the manuscript. Continuing with this example, if somebody strongly agree with something, at the same time agree; thus, although in the table the results are split in function of the questionnaire responses, in the text description it has to be used another way in order to avoid duplicate information.

Response: Thank you for the constructive comment. We corrected it throughout the manuscript. 

Comment 6: Table 3: “Uncerta Disagree in N(%) N (%)”. It should be corrected.

Response: Thank you. We have corrected (p,11. Line 250-251)

Comment 7: Please, replace “Percent” with “Percentage” in all tables.

Response : Thank you. We have corrected it. 

Comment 8: What do you mean with “Seventy-eight (34.8%) of the respondents did not know where to provide first aid”?

Response: it is to mean the site of our body to provide first aid (Heimlich man over or back slap) for complete airway obstruction. (p,15. Line 260-261). 

Comment 9: Table 5: Please, remove all data regarding no significant differences. “0.000” must not be used; it has to be replaced with “< 0.001”. CI lower and upper bounds should be separated by “-“. In addition, results of Table 5 should be revised; there are high differences between COR and AOR, is this correct? For example, in the case of Age (25-29), 0.156 becomes in 5.889.

Response: Thank you for your insight comment and advise. We have modified and tried to describe these in the manuscript (p,14. Table 5) 

Discussion:

Comment 1: Discussion should be start summarizing the main findings. Then, it should be follow the same order than results. The association between first aid training and knowledge was the last result reported, but it the first result discussed. Please, be consistent in this regard.

Response: Thank you for your guidance in comment. We have tried to correct based on comments accordingly. 

Comment 2: Again, the manuscript cited from China has no reference (l.237-238).

Response: We corrected it (p,15. Line 316)

Comment 3: L.238-239: It does not understand the following sentence: “But a study finding in Egypt was higher than our finding where majority of teachers were knowledgeable towards first aid management of chocking”

Response: It is to mean that teachers in Egypt were more knowledgeable compared with this finding. (p,15. Line 304-306). 

Comment 4: Discussion should address how to achieve that KG teachers train in first aid matters as choking management; for example, including this contents in university degrees, which was recommended in previous publications

Response: Thank you your comment and advice. We incorporate it in the manuscript in the discussion and recommendation part as first aid training to be included in the teachers’ educational curriculum.(p,16. Line 344-346) 

Comment 5: The following references should be to take into account to complement the introduction and the discussion:

https://doi.org/doi:10.1016/j.anpede.2018.10.013

https://doi.org/10.1097/EJA.0000000000001272

https://doi.org/10.1007/s00431-021-03971-x

https://doi.org/10.1016/j.resuscitation.2020.04.021

https://doi.org/10.1016/j.anpede.2019.10.005

Response: Thank you for suggesting us to use references. We have used these articles both in the introduction and discussion part in the manuscript. We greatly appreciate all reviewers’ thoughtful review and recommendations in the manuscript. 

While revising your submission, please upload your figure files to the Preflight Analysis and Conversion Engine (PACE) digital diagnostic tool, https://pacev2.apexcovantage.com/. PACE helps ensure that figures meet PLOS requirements. To use PACE, you must first register as a user. Registration is free. Then, login and navigate to the UPLOAD tab, where you will find detailed instructions on how to use the tool. If you encounter any issues or have any questions when using PACE, please email PLOS at figures@plos.org. Please note that Supporting Information files do not need this step

Response: We have registered with PACE and the figures meets PLOS requirements. We downloaded from the PACE and uploaded as Figure 1. 

Additional clarifications 

In addition to the above comments, spelling and grammatical errors pointed out by the reviewers have been tried to correct. 

We look forward to hear from you in due time regarding our submission and to respond to any further questions and comments you may have. 

Sincerely, 

Andualem Wubetie Aniley _____________

11/07/2021.

---

## [Editor Report · Decision Letter 3]

15 Jul 2021

Assessment of knowledge, attitude and practice on first aid management of choking and associated factors among kindergarten teachers in Addis Ababa Governmental Schools, Addis Ababa, Ethiopia.     A cross-sectional institution-based study

PONE-D-20-32556R3

Dear Dr. Mr Andualem Wubetie,

We’re pleased to inform you that your manuscript has been judged scientifically suitable for publication and will be formally accepted for publication once it meets all outstanding technical requirements.

Kind regards,

Sergio García López, Ph.D.

Guest Editor

PLOS ONE

Additional Editor Comments (optional):

Dear authors,

After reviewing the latest version provided by you, I am pleased to inform you that you have addressed all the considerations raised.

I appreciate the work done to improve the manuscript. This will facilitate its dissemination and scientific rigor.
---

## [Editor Report · Acceptance letter]

22 Jul 2021

PONE-D-20-32556R3 

Assessment of knowledge, attitude and practice on first aid management of choking and associated factors among kindergarten teachers in Addis Ababa governmental schools, Addis Ababa, Ethiopia.     A cross-sectional institution-based study 

Dear Dr. Wubetie Aniley:

I'm pleased to inform you that your manuscript has been deemed suitable for publication in PLOS ONE. Congratulations! Your manuscript is now with our production department. 

Kind regards, 

on behalf of

Dr. Sergio García López 

Guest Editor

PLOS ONE